# Perspective on Nanofiber Electrochemical Sensors: Design of Relative Selectivity Experiments

**DOI:** 10.3390/polym13213706

**Published:** 2021-10-27

**Authors:** Stanley G. Feeney, Joelle M. J. LaFreniere, Jeffrey Mark Halpern

**Affiliations:** 1Department of Chemical Engineering, University of New Hampshire, Durham, 03824 NH, USA; sgf1015@wildcats.unh.edu; 2Department of Mechanical Engineering, University of New Hampshire, Durham, 03824 NH, USA; jml1148@wildcats.unh.edu

**Keywords:** nanofibers, electrochemical sensing, selectivity experiments, biosensors, chemical sensors

## Abstract

The use of nanofibers creates the ability for non-enzymatic sensing in various applications and greatly improves the sensitivity, speed, and accuracy of electrochemical sensors for a wide variety of analytes. The high surface area to volume ratio of the fibers as well as their high porosity, even when compared to other common nanostructures, allows for enhanced electrocatalytic, adsorptive, and analyte-specific recognition mechanisms. Nanofibers have the potential to rival and replace materials used in electrochemical sensing. As more types of nanofibers are developed and tested for new applications, more consistent and refined selectivity experiments are needed. We applied this idea in a review of interferant control experiments and real sample analyses. The goal of this review is to provide guidelines for acceptable nanofiber sensor selectivity experiments with considerations for electrocatalytic, adsorptive, and analyte-specific recognition mechanisms. The intended presented review and guidelines will be of particular use to junior researchers designing their first control experiments, but could be used as a reference for anyone designing selectivity experiments for non-enzymatic sensors including nanofibers. We indicate the importance of testing both interferants in complex media and mechanistic interferants in the selectivity analysis of newly developed nanofiber sensor surfaces.

## 1. Introduction

Nanomaterials are known to exhibit unique mechanical, chemical, electrical, and optical characteristics when compared to more general materials [1]. This is due to their high aspect ratio and potential quantum effects at low dimensions. Nanofibers are 1D nanomaterials, meaning they are in the nanoscale range for two dimensions and are analogous to long threads. They have many of the same unique characteristics as other nanomaterials, but specifically have very large surface areas and porosities that lend well to adsorptive mechanisms with high mass transfer rates. Due to the fact that nanofibers have one macroscale dimension, their composition can contain a large number of functional groups in comparison to 0D nanomaterials like quantum dots, nanoparticles, or nanorods. As a result, nanofibers have a higher number of binding sites compared to their 0D counterparts. Additionally, when compared to 2D materials, 1D nanomaterials have higher surface area to volume ratios and temperature/temporal stability [2]. Nanofibers benefit from the ability to be fabricated through the electrohydrodynamic technique of electrospinning [3,4]. Electrospinning is an easy, low cost fabrication method that allows for variable morphology that is not applicable to other nanomaterials [5,6,7]. Electrospun nanofiber polymers such as polyaniline (PANI) can be less expensive than traditional catalytic nanomaterials [8,9,10,11,12]. Nanofibers have been investigated for use in a wide range of fields, including biomedical hydrogels, textiles, photovoltaics, pharmaceutics, water treatment, catalysis, optical computing, and sensors [13,14,15,16,17,18,19]. This review will focus on the use of nanofibers in sensing applications.

Nanofibers have high mass transfer rates and adsorption characteristics, which lead to higher sensitivity, lower detection limits, and greater temporal resolution in sensing applications [20,21,22]. Further, the high surface area to volume ratio and porosity of nanofibers increases the available analyte binding sites and molecular-surface interactions [23,24,25,26]. Graphene nanofibers, through careful adjustments in their structure, can be superior in biosensing to carbon nanotubes (CNTs) [27,28,29,30] as well as less expensive in general [5,6] due to high adsorption from their adjustable porosity and surface area. Nanofibers formed from long peptide chains, such as elastin-like polypeptides and elastin-like peptide amphiphiles, have also been explored for their stimuli responsive and adsorptive behavior [31,32,33,34]. In short, the physical properties of nanofibers demonstrate high favorability in sensor applications over traditional nanomaterials. 11 examples are summarized with lower detection limits (LDL) and precision values, reported as relative standard deviation (RSD), in Table 1.

Nanofibers have been previously reviewed for their sensitivity [46,47]. A variety of nanofiber materials are included in these sensors, including both organic and inorganic compositions, carbon nanofibers (CNFs) (e.g., graphite, graphene, CNTs), and peptide nanofiber (PNF) sensors. Composite nanofiber sensors, which use multiple different fiber materials on one surface, have also been used for sensing [48,49,50]. Commonly missing from previous reviews are discussions of selectivity in the context of the sensing mechanism. Descriptions of sensible design of selectivity experiments, in the context of sensing mechanisms, would be particularly useful to junior investigators designing their first sensor experiments. Therefore, using the 11 examples from Table 1, we looked into common nanofiber sensing mechanisms (electrocatalytic, adsorptive, and biorecognition) and how the mechanism influences necessary control experiments in the evaluation of selectivity.

## 2. Nanofiber Roles in Sensing Mechanisms

Nanofibers in the context of electrochemical sensing participate in a variety of mechanisms and modalities to improve sensor performance, including electrocatalysis, adsorption, and analyte-specific recognition. Optical sensing methods such as high performance liquid chromatography (HPLC) and surface-enhanced Raman spectroscopy (SERS) require the use of expensive equipment and high maintenance costs [51]. Both SERS and HPLC can detect adsorption and analyte-specific recognition non-destructively, but cannot perform the electrocatalytic detection of analytes non-destructively [52,53]. Nanofibers can also be used in electronic type sensors (e.g., field effect transistors, chemiresistors); however, these types of sensors are subject to hysteretic effects that can lead to complex data analysis [54,55,56]. Electrochemical sensing does not require expensive equipment, and is of particular interest because of the rapid transduction times, low hysteretic effects, and low transduction costs associated with sensing [51,57]. Figure 1 is a graphical representation of these three sensing mechanisms with specific sensor examples summarized in Table 1. To classify as an electrocatalytic nanofiber sensor, the nanofiber surface complex must have an active role in an electrocatalytic response. An adsorptive classification indicates that the sensor’s response occurs through a non-specific adsorptive nanofiber property enhanced by an increase in surface area. Analyte-specific chemical recognition indicates that the sensor uses some form of recognition element that has a specific binding event with the analyte. These mechanisms can be taken advantage of independently or in conjunction with each other in the design and construction of sensors; for example, adsorptive and electrocatalytic mechanisms are frequently used together. The sensing mechanism can depend on the nanofiber material. For example, metal oxides are efficient electrocatalysts for a wide variety of reactions; thus, metal oxide nanofibers are mainly useful for catalytic mechanistic sensing [58]. On the other hand, while organic polymeric nanofibers do not have the same electrocatalytic behavior, they still retain adsorptive characteristics and are easier to use because they don’t require heat sintering for adhesion to surfaces [59]. Peptide nanofiber materials are particularly useful for biorecognition mechanisms because of the various amino acid sequences that can be used for molecular imprinting or controlled conjugation of functional groups [60,61]. Often, the fabrication of the nanofiber can further influence the sensing mechanism. Therefore, a limited amount of fabrication information is included with each example to signify the influence of the fabrication process on the sensing mechanism. 

### 2.1. Electrocatalytic Activity

When a nanofiber material, or any other material, shows catalytic activity for an analyte of interest, the nanofiber material has potential to be used as an electrocatalytic sensor for that analyte. Due to the uniquely electroactive characteristics of nanofibers, derived from their large and adjustable active surface area, there is potential for their use in a variety of electrocatalytic applications [62,63,64,65,66,67]. The electrocatalytic behavior of nanofibers can also be tuned by altering their size or configuring their structure, changing the sensitivity and selectivity towards a target analyte [68,69,70]. 

As an example, Figure 2 depicts the electrocatalysis of electrospun RuO_x_-doped CeO_2_ nanofibers to the oxidation of carbon monoxide, as investigated by Liu et al. [62]. This depiction shows how the lattice structure of the CeO_2_ nanofibers aligns with and without the Ru-doping. Through N_2_ physisorption and the lattice parameters, it was found that the surface area of RuO_2_ by itself is approximately 20 m^2^/g, while the surface area in the CeO_2_ nanofiber lattice configuration is over 5 times that, at 117 m^2^/g. The catalyzed reaction with RuO_2_ had an oxygen consumption of 130 μmol/g, while the Ru-doped CeO_2_ nanofibers had a consumption rate over 10 times higher, at 1772 μmol/g. The specific mechanism for oxidation of CO by the CeO_x_ lattice is also depicted. CO is adsorbed to a clean CeO_x_ surface and oxygen is lent to the reaction by the lattice, forming an oxygen vacancy on the surface. Once the CO_2_ leaves the surface, the vacancy allows for the adsorption of O_2_. With the presence of an extra oxygen atom on the surface, a second CO molecule adsorbs and subsequently leaves as CO_2_, fully regenerating the clean CeO_x_ surface.

While Figure 2 depicts one modality of electrocatalytic activity, there are other modalities for eliciting electrocatalytic behavior. Four additional examples of electrocatalytic sensing modalities are detailed: Metal oxide nanofibers can directly act as electron carriers as the source of a sensor’s electrocatalytic behavior [36,37], functioning largely as class-recognition type sensors (Table 1, #2 and 3). In the case of one biosensor for purine detection (Table 1, #2), CuO nanofibers (CuO NFs) and ZnO nanoparticles (ZnO NPs) were immobilized within a poly-L-cysteine (PLC) matrix [36]. The surface was prepared through simultaneous electropolymerization in a buffered, aqueous solution of L-cysteine (LC), ZnO NPs, and CuO NFs. The CuO-ZnO heterostructures were said to form p-n junctions that greatly enhanced sensitivity. The sensitivity of the sensor increased from 0.353 to 2.66 μA/μM for guanine and from 0.155 to 2.67 μA/μM for adenine when compared to a sensor that uses metal nanoparticles without nanofibers. This improvement was attributed to a synergistic combination of the electrocatalytic behaviors of the two metal oxide nanostructures. The nanofibers also increased the electron transfer capacity of the electrode surface.N-doped carbon nanofibers (NCNF) can be used in conjunction with N-doped graphene quantum dots (NGQD), as both NCNFs and NGQDs have catalytic activity toward nitrite and the combination of the two produces much higher sensitivity (Table 1, #7) [41]. The composite with NGQDs was formed by hydrothermal treatment with in situ quantum dot synthesis. The dried NGQD/NCNF was physically adhered to the surface of glassy carbon electrodes. Introduction of the doped nanofibers into the glassy carbon electrode resulted in an increase in the oxidation peak current from 9.914 to 20.56 μA. For nitrite sensing, The NGQD/NCNF composite biosensor showed an improvement in the limit of detection from 8.1 to 3 μM when compared to a porous graphite sensor. The doping of the CNFs with heteroatoms, in this case nitrogen, synergistically working with NGQDs by increasing electron transfer, allowed the fibers to electrocatalyze the oxidation of nitrite.An example of a sensor that uses complex surface interactions to achieve redox catalysis is a polymethylene blue (PMB)-decorated Cu-CNF sensor for the oxidation of creatinine (Table 1, #9) [43]. An activated carbon microfiber (ACF) surface was dispersed with Cu(NO_3_)_2_ which was used for in situ Cu NP synthesis. CNFs were added to the surface via chemical vapor deposition. The PMB was then synthesized on the surface through an electro-polymerization method. The full construction of the surface was a PMB nanofiber matrix on-top of a copper infused CNF/activated carbon surface mat (PMB-Cu-NF/ACF). The CNF/ACF mat is an adsorptive surface, but the interactions between the PMB-Cu promotes the selective catalysis of creatinine oxidation. Detection, with differential pulse voltammetry and cyclic voltammetry, resulted in an improvement of the detection limit from 56.55 to 0.24 ng/mL when compared to a copper electrode without the PMB NFs or CNFs. This improvement was attributed to the selective nature of the surface interactions promoted by the electrocatalytic activity promoted by the large surface area of the nanofiber matrix.Carbon-based nanofibers can be fabricated to have a high density of states for an increased electrocatalytic response. One sensor was fabricated by carbonizing electrospun poly-acrylonitrile onto graphitized fiber carbon paper. The high density of electronic states led to a wide detection range of dopamine oxidation. [71]. The deposition time was varied from 5 h to 33 h, which lead to the dynamic range increasing from 8–9000 μM to 0.2–700,000 μM. Additionally, the limit of detection improved from 5.58 to 0.07 μM. The dynamic range and the limit of detection were improved by orders of magnitude due to the high surface area and edge effects of the nanofibers.

### 2.2. Adsorptive Behavior

Nanofibers can be used as an adsorbent for a variety of analytes due to the nanofiber’s highly adjustable porosity and surface effects [72,73,74,75,76]. As with the electrocatalytic behavior, the adsorptive properties of nanofibers can be adjusted by changing the size, configuration, and synthesis methods. The adsorptive property of nanofibers can also be used in the removal of ions from wastewater, and therefore, many modern nanofiber sensors allude to their high adsorbent properties as a key feature in their performance. The adsorption mechanism mostly contributes to other sensing modalities, like electrocatalysis or analyte-specific recognition, to improve overall sensor performance. As an example, Figure 3 depicts the adsorptive behavior of Cu-doped boron nitride nanofibers to carbon dioxide [75], where nitrogen vacancies, as well as the addition of doping agents, can change the energy of carbon dioxide adsorption to the nanofibers. The fiber matrix was fabricated by chemical synthesis with Cu seed crystals. The energy of adsorption (E_ads_) of the matrix (Figure 3) indicates the adsorptive capability of the boron nitride nanofiber matrix, which increased from 0.239 to 0.711 eV after introducing nitrogen vacancies and Cu clusters. The Cu-doped boron nitride nanofibers were found to have an adsorptive capacity of 2.77 mmol/L, higher than that of other comparable systems, while still having near 100% reusability. Incorporating both a nitrogen vacancy to adjust porosity and Cu clusters to promote specificity leads to a more adsorptive and reusable surface. The tunability of the adsorptive properties would lead to flexibility in sensor properties depending on the specific adsorptive and sensitivity needs.

Adsorptive effects can enhance the limit of detection and the sensitivity of sensors, as is the case of the metallic and metal oxide sensors. Because of their enhanced adsorption, nanofiber-based materials have potential for use in sensitive, accurate, fast, and precise electrochemical sensors. Four additional examples of adsorptive mechanisms are detailed:An atrazine sensor that uses tin (IV) oxide nanofibers to achieve a very low limit of detection by using nanofiber adsorption to enhance atrazine interaction with traditional antibody biorecognition elements was created (Table 1, #10) [44]. Atrazine antibodies were grafted to SnO_2_ NFs that were predispersed onto a glassy carbon electrode surface. The antibodies promoted specific interaction via traditional biorecognition routes. The absorptive effects of the tin (IV) oxide promoted a lower limit of detection (0.9 zM) which is orders of magnitude smaller than other atrazine sensors (typically around 20 pM) [44].Organic polymer nanofibers do not have the same electrocatalytic potential as metal oxides, but due to their tunable size and inexpensive fabrication, are commonly used for their adsorptive effects in combination with electrocatalytic compounds (Table 1, #4, 5, 6, and 8) [38,39,40,42]. A specific example of this is a composite sensor made with polypyrrole (PPy) NFs for the simultaneous electrocatalytic determination of ascorbic acid, dopamine, paracetamol, and tryptophan (Table 1, #4) [38]. ZnO nanosheets and Cu_x_O nanoparticles were electrochemically deposited on PPy NFs to create a 3D CuxO-ZnO NP/PPyNF/RGO structure. The zinc oxide–copper oxide p-n junction heterostructures electrocatalytically oxidize the analytes. The PPy NFs were used to increase the adsorption of the analytes to the surface, which increases sensitivity, as well as to prevent graphene sheet aggregation for an increase in stability. An increase in the linear range from 0.5–20 μM to 0.04–420 μM of dopamine and a decrease the in limit of detection from 0.17 to 0.012 μM of dopamine was observed compared to Ni and CuO modified surfaces without the nanofiber.In the aforementioned creatinine sensor, the PMB fibers produce a catalytic effect while the copper dispersed CNF composite promotes adsorption to improve sensor performance. (Table 1, #9) [43]. This is an example of combining two different nanofibers in such a way that they have separate but complementary roles. The CNFs used in this sensor increase the adsorption of creatinine to the surface, resulting in an additional increase of reported sensitivity.Another example of a sensor that uses the adsorptive mechanism of nanofibers is a pH and H_2_O_2_ sensor that uses a layer-by-layer assembly of PAA/PANI nanofibers [77]. The PANI nanofibers were synthesized using ammonium persulfate chemistry and were deposited onto a cleaned glassy carbon electrode in alternating fashion with PAA. The numbers of layers of PAA and PANI resulted in different adsorptive properties, and therefore, different electrochemical response. After six layers of PAA and PANI, the linear range of the sensor increased from 0.005–0.8 to 0.001–6 mM and the detection limit improved from 1.2 to 0.3 μM. The improvement of these properties was attributed to the high surface area and microporosity of the sensor surface, which can be tuned by changing the modification procedure.

### 2.3. Analyte-Specific Recognition

One of the reasons the electrocatalytic and adsorptive properties of nanofibers are so frequently exploited for sensor design is that they allow for non-enzymatic sensing, which avoids many of the drawbacks of classical chemical recognition elements [78,79,80]. However, electrocatalysis and adsorption sensing mechanisms lend themselves to non-specific, class-recognition sensing rather than analyte-specific sensing, which can result in selectivity issues. One way to mitigate these selectivity issues is to add analyte-specific interactions into the nanofiber surface structure. Traditional chemical recognition elements, such as an antibody or enzyme, can be combined with adsorptive nanofibers to enhance an interaction with the analyte of interest [29,81,82,83], but also tend to be costly, have stability issues, and have limited detection ranges [84,85,86,87].

Some nanofiber-based sensors, such as peptide nanofibers (PNF) or other organic polymer nanofibers, make use of analyte-specific affinity interactions to avoid the drawbacks of antibodies and enzymes while maintaining high specificity [43,45]. Most analyte-specific recognition elements are used to increase the specificity of the sensor, but may also increase the sensitivity as well. Figure 4 shows an example of an analyte-specific binding event using a self-assembled PNF that is able to selectively self-assemble onto a lymphocyte receptor, CD44, to electrochemically detect breast cancer stem-like cells (BCSCs) (Table 1, #11) [45]. Cancer cells, which express CD44, bind to the surface; a traditional biorecognition element, with nucleolin AS1411, is attached to the surface to capture BCSCs. The PNF is designed with specific functional groups with distinct CD44 binding sites. PNF is added to the solution and selectively binds to BCSC affixed to the surface via the CD44 receptors. Afterwards, silver nanoparticles (Ag NPs) are selectively modified to the PNF attached to the surface-bound CD44; the BCSC/PNF/AgNPs generate a unique and selective electrochemical signal. This kind of construct can also be thought of as a multifunctional nanofiber. Because of the pseudo sandwich assay construct of the sensor, there are multiple analyte-specific recognition elements present in the sensor: (1) the specific nucleolin:BCSC construct, (2) the design of the PNF to selectively bind to the BCSC surface, and (3) the Ag NPs that selectively modify to the PNF bound on top of BCSCs that generate a unique electrochemical signal. The construct was monitored via electrical impedance spectroscopy and linear sweep voltammetry. This is another case of non-classical biorecognition elements in the form of a peptide-based aptamer-style modality for improvement of selectivity. When the sensor response to BCSCs was compared to that of interfering compounds, the most interference was caused by HepG2 cells with a 1.5% signal response, compared to the 12.5% signal response from BCSCs and 1% background signal.

In recent years, nanofiber sensors have been developed to incorporate analyte-specific interactions, in addition to the class-recognition modality of electrocatalytic enhancement and adsorptive mechanistic sensors. Three additional examples of analyte-specific interaction modalities are as follows:One type of analyte specific interaction occurs between traditional biorecognition elements like antibodies or enzymes. For example, the previously mentioned atrazine sensor uses a traditional biorecognition element, an antibody, integrated into a nanofiber network (Table 1, #10) [44]. The antibody increases the specificity from the specific analyte interaction, and the nanofiber network provides an increase in antibody loading (high surface area) and an increase in the binding kinetics via adsorption. The interfering compound that was found to interact with the sensor the most was melamine, with a 15.6% change in peak current at 1 μM, while atrazine at that same concentration resulted in a 43.5% change at the same concentration. Additionally, a 7.2% interference was found in a 1:1 mixture of atrazine and urea.Analyte-specific interactions can also take the form of highly selective chemical interactions with surface lattice structures. The previously mentioned creatinine sensor has a specific binding event between the PMB-Cu nanofiber matrix and creatinine, allowing the PMB-Cu heterostructure to act as a synthetic chemical recognition element (Table 1, #9). [43]. The sensor was tested for multiple interfering compounds in clinically relevant ratios, and some non-specific adsorption was found which could be mitigated by washing. The end result after washing was minimal interference. The sensor was also tested in cerebrospinal fluid, saliva, and blood serum, resulting in an average recovery of 98.3%.Specific steric repulsion can occur between elements of a sensor and potential interferents, as is the case between biofouling proteins and poly(ethylene) glycol (PEG), as seen in a polyaniline nanofiber-based DNA sensor (Table 1, #5) [39]. There is a specific steric repulsion between the PANI/PEG composite and potential biofouling proteins. DNA capture probes were attached to a PANI NF/PEG surface, which added an additional specific interaction that improved selectivity, an analyte-specific interaction between DNA capture probes and the DNA analyte. The improvement in selectivity was evaluated by comparing the response of the sensor to DNA with one base pair mismatch from the target in a 10,000-fold concentration. Even at such high concentrations, the mismatched DNA only produced 25% the signal compared to the target DNA.

## 3. Selectivity Experiments in the Development of Nanofiber Sensors

Nanofiber-based electrochemical sensors are commonly designed with a large active surface area for the detection of electrocatalytic activity, surface adsorption, or both modalities [46,88,89]. Due to the nature of many of these sensors, the mechanistic class-recognition catalytic sensing modality is often confused with analyte-specific sensing. The experimental design of selectivity tests and specific reporting language is critical to the development of inexpensive, rapid-response sensors. We intend to clarify the need for the appropriate design of selective experiments for both mechanistic class-recognition sensing modalities (i.e., adsorptive and electrocatalytic nanofiber sensors).

Selectivity experiments provide clarification of relevant interfering compounds and variables, but the methodology of the experiments must adequately reflect the intended sensor mechanism and application. While real complex-media sample tests are exceptionally valuable to understanding selectivity, they do not replace thorough interrogation of mechanistic-interfering compounds. Additionally, real complex-media sample tests must be an accurate reflection of the intended end-use for true relevancy in the robustness of the sensor. We examined the experimental design of selectivity tests for nanofiber-based electrochemical sensors and propose procedures to improve reporting.

### 3.1. Interferant Control Experiments

Control experiments to test for the selectivity of a sensor are commonly referred to as interferant control experiments; an important decision for sensor development is what compounds and variables to use as interferants. Interferant control experiments strengthen the conclusions, leading to a reliable sensor product [87]. Interferant control experiments typically occur by adding the identified potential interferant chemical to a buffered solution. In general, for any sensor, commonly chosen interfering compounds are based on both the expected sensor environment and the different interferant sources. The choice of these compounds is particularly important in a mechanistic class-recognition modality (i.e., electrocatalytic or adsorptive) rather than analyte-specific detection. The classes of compounds tested should cover a wide range, including anionic and cationic compounds, organic species that might be present in a real sample, compounds with similar functional groups to the analyte, and any compound that may undergo the same mechanistic interaction as the analyte [35,36,43]. The relative concentrations of interfering compounds to the tested analyte should be equal to or greater than what is typically found in the intended real complex-media sample. A sensor that has a small signal response in the presence of interfering compounds (i.e., at interferant concentrations well above relevant use) indicates good sensor selectivity [38,39,41,44]. Table 2 expands on Table 1’s examples by detailing the interferant compounds and variables tested and the highest found interferant; the highest found interferant is the interferant that lead to the largest deviation from sensor calibration by reporting the percent deviation (if available).

**Table 2 polymers-13-03706-t002:** Summary of control interferant experiment data for various nanofiber-based electrochemical sensors. The # corresponds to the same designation number in Table 1.

#	Analyte	Tested Interferant Compounds	Highest Found Interferant	Ref.
1	Idarubicin hydrochloride	Ca^2+^, Mg^2+^, Fe^2+^, Cl^−^, glucose, lactose, fructose, AA, CA, UA, urea, acetaminophen, epirubicin, doxorubicin, daunorubicin, cysteine	3% from 5-fold cysteine and AA	[35]
2	Adenine, guanine	Na^+^, Mg^2+^, Ca^2+^, Cu^2+^, Zn^2+^, Fe^3+^, CO_3_^2+^, NO_3_^−^, Cl^−^, thymine, xanthine, cytosine, tyrosine, tryptophan, aspartic acid, pyridoxine, AA, FA, UA, glucose, alanine, glycine, arginine, L-cysteine	17.6% from 200-fold tryptophan toward guanine determination 9.0% from 500-fold tryptofan toward adenine determination	[36]
3	Acetaminophen	UA, DA, AA, glucose	DA *	[37]
4	AA, DA, Paracetamol, and tryptophan	Cytesine, epinephrine, glucose, UA, FA, and tyrosine	6.96% from 500-fold FA	[38]
5	DNA sequence	BSA, HSA, IgG, Hb, base-mismatched DNA	25% from 10,000-fold single base-pair mismatched DNA	[39]
6	Hydrogen peroxide	AA, UA, DA	1-fold AA, UA, and DA *	[40]
7	Nitrite	K^+^, Ca^2+^, Na^+^, Mg^2+^, Zn^2+^, Ag^+^, NH_4_^+^, Cl^−^, NO_3_^−^, CO_3_^2−^, HCO_3_^−^, PO_4_^3−^	10% from 100-fold Ag^+^ and Zn^2+^	[41]
8	H_2_O_2_, glucose	AA, UA, DA	AA *	[42]
9	Creatinine	DA, AA, UA, cholesterol, urea, glucose, glutamine, bilirubinketones, hemoglobin, pyruvic acid	Clinically relevant ratios of all compounds *	[43]
10	Atrazine	Urea, glucose, antibiotic, BSA, HSA, Na^+^, melamine	15.6% from 1-fold melamine	[44]
11	BCSC	BT-474, HepG2, L02	1.5% from 1-fold HepG2	[45]

* % of interference not reported.

As a case study, a carbon/polymer composite nanofiber-based sensor was used for the simultaneous detection of lead and cadmium in wastewater runoff; 10 different metal ions and two anions (SO_4_^2−^ and NO_3_^2−^) were tested within the interferant control experiments [90]. The sensor’s intended purpose, ionic content within wastewater runoff detection, would have other ions (metal and anions) which are predicted to be common interferants; these predicted common interferants were properly tested. Additionally, interferant control experiments should be conducted in solutions at, or well above, the appropriate concentrations relative to the analyte of interest. Interferants at concentrations 500-fold above expected levels were tested within this case study [90].

While interferant control experiments are performed, we have found authors should additionally consider mechanistic interferants defined as interfering molecules that can potentially be detected through, or otherwise interfere with, the mechanistic class-recognition modality of the sensor. As an example of these types of mechanistic interference, a nanofiber-based glucose sensor for use in saliva appropriately tested nine different interfering compounds that would be commonly found in biofluids. The sensor used a composite surface of reduced graphene oxide, CNFs, and CuO nanoneedles to electrocatalytically detect the oxidation of glucose. [91]. Each of the tested interfering compounds, including uric acid (UA), ascorbic acid (AA), KCl, NaOH, dopamine (DA), acetaminophen, galactose, and lactose, have the potential to act as a mechanistic interferant in a different way, as shown in Figure 5:

Electrochemical oxidation peaks have the potential to be misinterpreted when the equilibrium potential of interferants is similar to that of the target analyte. For example, the peaks of AA, UA, DA, and acetaminophen are similar to the oxidation peaks of catalyzed glucose (Figure 4), and can interfere with the interpretation of the data [91]. When a sensor is run potentiometrically, the oxidation and reduction potentials should be investigated in addition to the signal strength. Typically, this will affect interferants with similar electrochemically active functional groups in electrocatalytic modalities, as seen in Table 2 (#2, 3, 4, 6, and 8). Within these examples, the oxidative or reductive overlap was found to have the largest interference.Common functional groups between the molecules have the potential to adsorb similar to the target analyte. Galactose and lactose were tested because, as with glucose, they are sugars with a similar chemical structure (Figure 4) [91]. Similar chemical structures are a commonly tested interferant. Testing the adsorption of similar chemical structures is very important in adsorption or analyte-specific recognition sensors. Adsorptive modality disruptions are more common when a sensor runs via an amperometric or impedimetric modality. As seen in Table 2 (#1, 5, 10, and 11), the primary interferant is one of a similar chemical structure.Surface interactions through charge-induced adsorption, such as ions, can alter the surface electrolyte double-layer capacitance and influence the electrochemical output. These induced changes can result in an artificially higher or lower concentration measurement via alteration in the electrochemical transduction output. For example, KCl can change the overall charge of the solution, and small changes can result in an altered signal (Figure 4) [91]. Also, NaOH has the potential to lower the acidity of the analyte solution, which means that changes in water ionic charges should be tested as an interferant as well. Ionic interference was only tested in three examples (Table 2, #1, 2, and 7), but was found to be the major interferant in example #7 (Table 2). We would expect ionic changes to be more commonly found to interfere with electrochemical transduction if these interferant control experiments were more widely tested.

While they were not tested for in the study referenced in Figure 4 [91], other mechanistic interferants to consider are interfering variables such as temperature and viscosity. Temperature, viscosity, and other thermodynamic variables can affect the chemical potential at a sensor’s surface and can potentially influence all electrochemical transduction techniques and mechanistic modalities. Further, these thermodynamic variables are rarely tested, and we saw no reference to them in the examples provided in Table 2. While these variables are usually controlled by the choice of sample and testing conditions, we suggest researchers stay conscious of these issues. One possibility is a small sensitivity analysis of these variables to determine how much sample and equipment control is needed in non-tested interferants.

### 3.2. Transitioning to Real Samples and Analysis

Real complex-media sample analyses are necessary and useful to test the robustness and capabilities of a nanofiber-based sensor and to determine sample pretreatment needs. Often the non-specificity or class-recognition action of nanofibers are underreported or underrepresented; changes in other chemicals within the complex media can lead to false positive or negative measurements if not properly tested [71,87]. While interferant control experiments are usually carried out in a purified solvent with added interferants, real complex-media samples (or intended end-use media) can change the sensor performance and analyte response [92,93,94]. These changes can be the result of viscosity effects, the presence of multiple interferants at once, temperature dependencies, or other unpredictable variables. As an example, a metal/polymer/graphene nanofiber was used to sense glucose via a combination adsorption/electrocatalytic mechanism, and the blood sample was diluted with buffer to reduce viscosity-based changes, leading to increased consistency via sample preparation [94]. Overall, a real sample analysis provides extra context for future use of the sensor and supports the efficacy of the design.

Both the choice of media and pretreatment can influence the catalytic or adsorptive mechanism of the sensor, leading to additional (or reduced) interference, and is critical in accurate sensor reporting. Two of the sensors did not undergo real complex-media analysis, both of which are adsorption-based (Table 3, #6 and 8). The choice of complex media must always reflect the desired end-use application. Pretreatments must also be carefully considered to improve the robustness, selectivity, and reproducibility of measurements. When making comparisons of the real complex-media analyses of the nine examples from Table 1 that tested complex media, four of the sensors reported very little sample preparation (one or less steps), indicating a design for simplified end-use [37,38,39,43,44,45]. Of these, three used some form of analyte specific recognition [39,43,44,45]. One of the samples that requires extensive pretreatment [35,36,41] is a nitrite sensor for tap water or food testing [41], where the pretreatment would be unlikely to create complications for the intended application (Table 3, #7). Overviews of the pre-treatment strategies of the reports in Table 3 are as follows:Dilution was a step during sample preparation for four of the reported sensors (Table 3, #1, 2, 4, and 7): the TiO_2_/CNF sensor for idarubicin hydrochloride [35]; the adenine and guanine sensor that uses a PLC/ZnO-NPs/CuO-NF modified surface [36]; the 3DCu_x_O-ZnO NP/PPyNF/RGO sensor for simultaneous detection of ascorbic acid, dopamine, paracetamol, and tryptophan [38]; and the NGQD/NCNF sensor for nitrite determination [41]. Simple dilution is relatively easy to perform, and therefore, would be acceptable even for untrained individuals. Dilution is primarily performed to reduce non-specific binding and physisorption of interfering agents in the complex media. It can also help to reduce viscous effects that would prevent efficient diffusion of the analyte to the surface. The detailed sensors that use this sample preparation (Table 3, #1, 2, 4, and 7) have highly adsorbent surfaces that benefit from the dilution step to promote the binding of the analyte.Centrifugation and filtration were steps during the sample preparation for two of the sensors (Table 3, #1 and 7): the TiO_2_/CNF sensor for idarubicin hydrochloride [35] and the NGQD/NCNF sensor for nitrite determination [41]. Centrifugation and filtration, depending on the application, may require special equipment that would limit the point-of-need use of this type of sensor. Centrifugation and filtration are useful in separating large components like proteins, cells, and macromolecules from biological samples. The separation of large macromolecules is important for the idarubicin and nitrite sensors because the large molecules could non-specifically bind to and foul the surface interfering with the adsorption mechanism (Table 3, #1 and 7).Ultrasonication was a step in just one of the sensors (Table 3, #10): the SnO_2_ for the determination of atrazine [44]. Ultrasonication requires the use of specialized equipment such that point-of-need applications are rarely suitable. Ultrasonication is used to break larger molecules down into smaller parts, which minimizes the adsorption of potential interferants compared to the intended analyte. The atrazine sensor (Table 3, #10) uses antibodies as biorecognition elements, and the ultrasonication breaks down larger proteins that could be susceptible to non-specific binding.pH adjustment was a sample preparation step in the complex media analysis of the adenine and guanine sensor that uses a PLC/ZnO-NPs/CuO-NF modified surface (Table 3, #2) [36]. The pH adjustment should be performed by a trained individual to ensure the pH adjustment is properly achieved. As previously mentioned, the charge of a solution affects the double layer capacitance of a surface, and therefore, influences the adsorption of the analyte and interfering compounds. Attaining a favorable pH environment is necessary for the selectivity and sensitivity of some sensors. Further, the pH has to be adjusted occasionally to increase the accessibility of the analyte. For example, a DNA sensor (Table 3, #2) uses HCl to digest the DNA to increase the accessibility of adenine and guanine in complex media [36]. However, the designed sensor operates better at neutral pH, which further makes a pH adjustment with NaOH after digestion necessary.

As a case study, a graphene/gold nanofiber composite biosensor for bisphenol A, a common fresh water contaminant [95], used extensive pretreatment for the detection of extracted bisphenol A from bottles. The sample preparation involved cutting a baby bottle into small pieces, ultrasonication in chloroform, solvent extraction in sodium hydroxide three times, and then dilution. The accuracy, a recovery rate between 98.4% and 102.1%, was obtained in comparison to a high-performance liquid chromatography control, and the recovery rate did not vary with potential interferants or with any step in the pretreatment process. However, the detailed analysis could not be readily applied in point-of-need applications, and the approach would need to be reevaluated for different applications, such as direct water testing. In other words, the results do not necessarily translate to complex-media solutions or different sample preparation modalities. For comparison, revisiting a previous example, a lead and cadmium ion nanofiber sensor [90] directly used river water samples with an extensive sample preparation involving digesting with nitric acid, nitric acid acidification, evaporative separation, and HCl pH adjustment. While also inapplicable to other complex media or different pretreatment methods, this example specifically demonstrated end-user application in river water for accurate lead and cadmium measurements.

**Table 3 polymers-13-03706-t003:** Summary of the real sample analysis data for various nanofiber-based electrochemical sensors. The # corresponds to the same designation number in Table 1.

#	Complex Media Tested	Sample Preparation	Ref.
1	Human serum, human urine	Centrifugation, filtration, dilution	[35]
2	Sturgeon sperm DNA, Human blood DNA, Flavithermus DNA	Digestion in HCl, heating, rapid cooling, neutralization with NaOH, dilution in PBS	[36]
3	Human serum	None specified	[37]
4	Human serum	Dilution	[38]
5	Human serum	None specified	[39]
6	N/A	N/A	[40]
7	Sausage, pickle, lake Water, tap water	Sausage and pickle: deproteinization, centrifugation, filtration, dilution with PBS Water: centrifugation, filtration, dilution with PBS	[41]
8	N/A	N/A	[42]
9	Human serum, human Cerebral spinofluid, Human saliva	None specified	[43]
10	Ground water, river Water	Ultra-sonicated	[44]
11	Fetal bovine serum	None specified	[45]

N/A indicates wasn’t tested in complex media.

### 3.3. Future Perspectives on Selectivity Experiments

While interferant experiments and complex media analysis have separate roles within selectivity control experiments, they work synergistically to provide a larger picture of selectivity for a given sensor. Both interferant control experiments and complex media experiments should be completed and reported to promote future nanofiber sensing use for point-of-need applications. Interferant experiments aim to show that individual compounds with the potential to interfere with the sensor, especially through the sensor mechanism, do not have a significant impact on the readout even at high concentrations. Within the context of interferant control experiments, a sensitivity analysis with each of the potential interferants will indicate a greater robustness of sensor and mechanistic design. Complex media analysis, on the other hand, aims to replicate how a sensor will perform when applied to a real sample after pre-treatment methods are applied. Typically, the recovery experiments are conducted in complex media, with a spiked analyte in the complex media to determine the response with respect to non-complex media. However, the spiked complex media experiments tend to be digital in design, as in only measuring a single concentration. More analog information, or multiple concentrations, can provide additional sensitivity information, including updated lower detection limit values. Further, the combination of interferant control experiments in complex media would provide a more comprehensive selectivity analysis of a sensor compared to only separate experiments, as represented in Figure 6. In all cases of selectivity experiments, accurate and full representation of data and results is crucial. Moving forward, we suggest that all nanofiber sensors conduct both interferant experiments and complex media analyses with a full report of the methodology and data.

## 4. Conclusions

Nanofiber-based electrochemical sensors show great promise for sensitive, precise, and selective measurements in complex media. Advantageous sensor properties are a direct result of the large specific surface area and adjustable porosity of nanofibers which allow nanofibers to promote electrocatalytic and adsorptive mechanisms for electrochemical sensing. These mechanisms are commonly used by sensors like the ones presented throughout the article and in Table 1. The mechanistic class-recognition nature of nanofiber sensors leads to selectivity issues that can stall advancement in various sensor applications. The selectivity of nanofiber-based electrochemical sensors can be appropriately characterized and improved by the thoughtful inclusion of relevant controls, the use of well-designed real sample analyses, and the inclusion of analyte-specific interactions. The improvement of sensor performance by inclusion of analyte-specific interactions by multiple sensors [39,43,44,45] can be observed in the selectivity experiment review in Table 2. Relevant interference studies and complex media experiment controls can validate the selectivity of a given sensor as is represented by Figure 6. We challenge researchers to also think about mechanistic interference when testing potential unexpected chemical variants in evaluating the robustness of a sensor. The specific expected interferant is not always the chemical analog closest to the target analyte, but the interferant likely to mechanistically disrupt detection. Researchers should remain conscious in their experimental design based on the proposed mechanism in sensing.

## Figures and Tables

**Figure 1 polymers-13-03706-f001:**
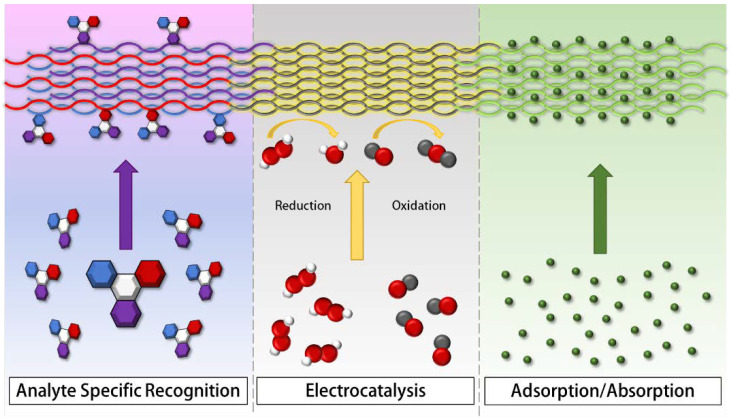
Diagram representing the primary sensing mechanisms for nanofiber-based sensors. **Left**: analyte specific recognition (ASR)—specificity is represented by the colors of the fiber matching the color of the functional group on the analyte it binds to. **Center**: electrocatalysis (EC)—reduction or oxidation is catalyzed by enhanced electron movement through nanofibers. **Right**: adsorption (AD)—molecules are trapped on nanofiber surfaces due to high porosity and large specific surface area.

**Figure 2 polymers-13-03706-f002:**
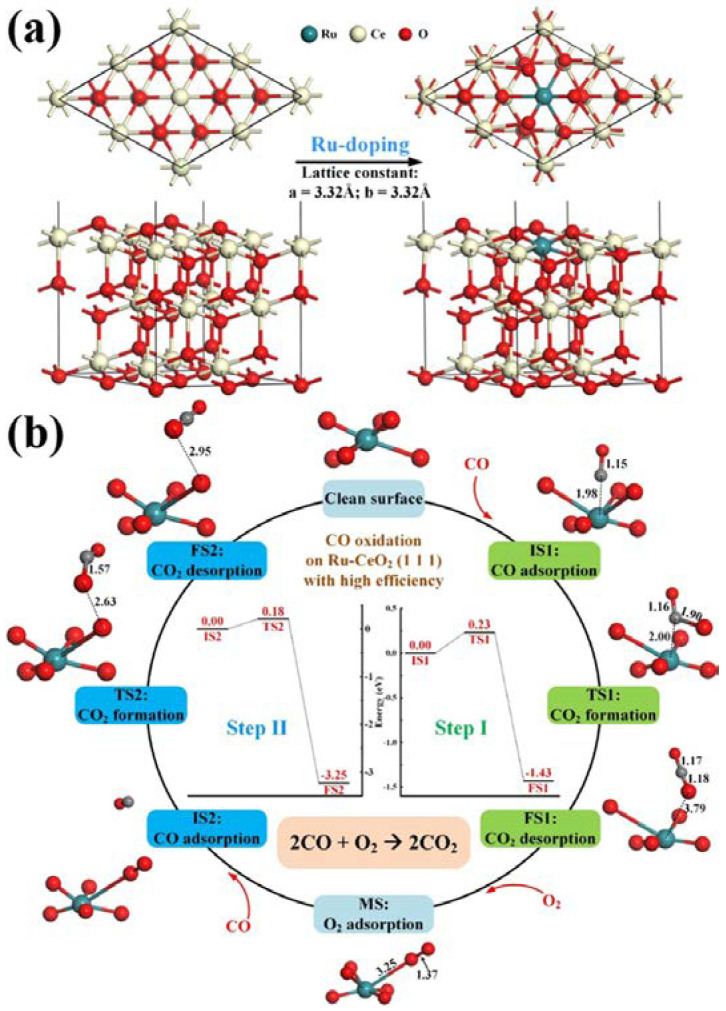
Mechanistic behavior of Ru-doped CeO_2_ as an adsorptive electrocatalyst for CO oxidation, as reported by Liu et al. (2020) (**a**) Diagram showing the Ru-doping process and lattice formation of the catalytic structure. (**b**) Diagram showing the reaction steps for the catalyzed carbon monoxide oxidation reaction. Reprinted with permission from *ACS Appl. Nano Mater.* 2020, 3, 8403–8413 [62]. Copyright 2020 American Chemical Society.

**Figure 3 polymers-13-03706-f003:**
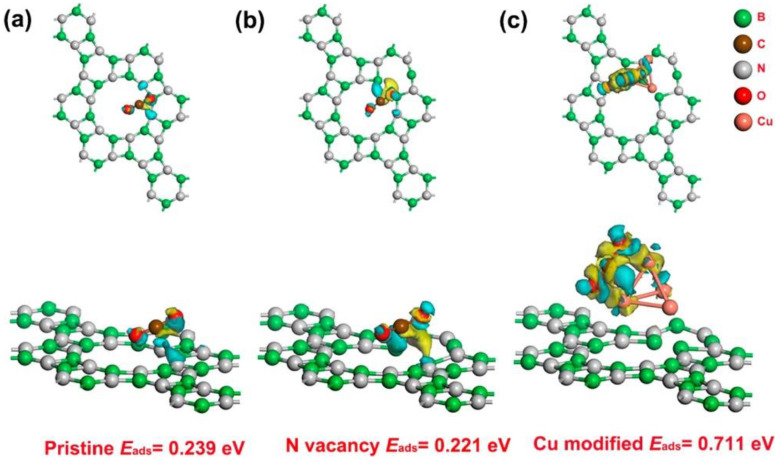
Theoretical adsorptive behavior of CO_2_ onto Cu-doped boron nitride nanofibers as reported by Liang et al. (2020). (**a**) Adsorption of CO_2_ onto pristine nanofibers. (**b**) Adsorption of CO_2_ onto nanofibers with nitrogen vacancies. (**c**) Adsorption of nanofibers onto nanofibers doped with Cu. Reprinted (adapted) with permission from *ACS Sustain. Chem. Eng.* 2020, 8, 7454–7462 [75]. Copyright 2020 American Chemical Society.

**Figure 4 polymers-13-03706-f004:**
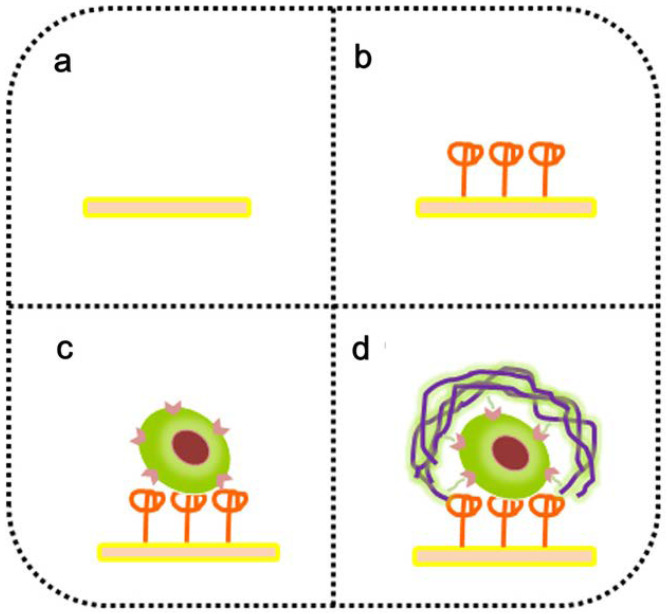
Diagram showing the stages of sensing CD44 on the surface of breast cancer stem cells as explored by Tang et al. (2019). (**a**) Bare gold electrode, (**b**) gold electrode with nucleolin AS1411, (**c**) attachment of breast cancer stem-like cell containing CD44, and (**d**) analyte-specific interaction between the functional groups of the multi-functionalized PNFs and the surface CD44 molecules. Reprinted (adapted) with permission from *Anal. Chem.*, 2019, 91, 7531–7537 [45]. Copyright 2019 American Chemical Society.

**Figure 5 polymers-13-03706-f005:**
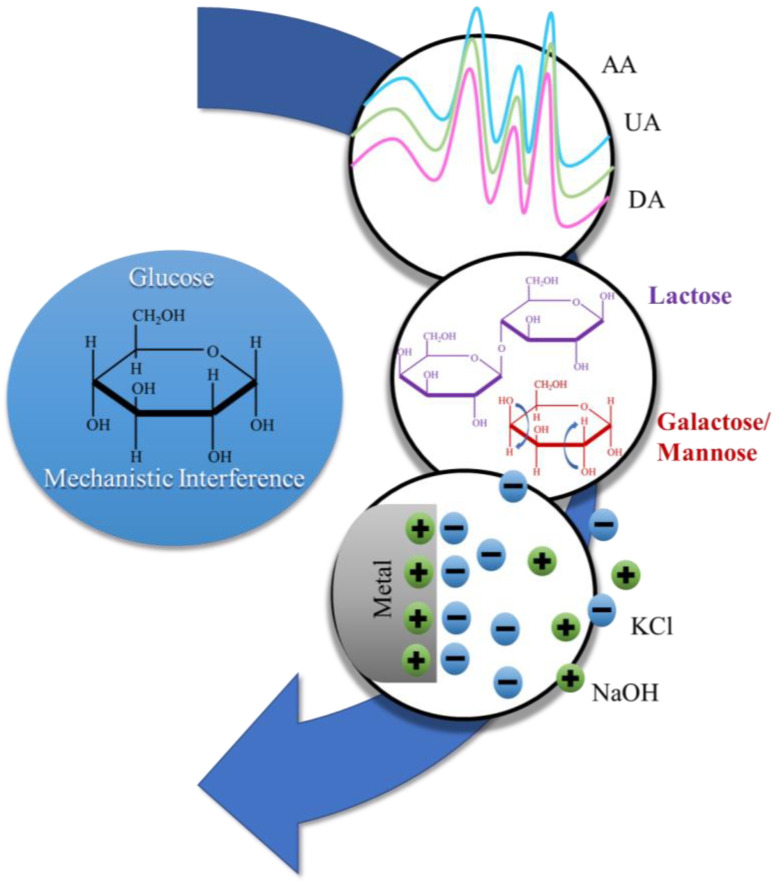
Graphic representation of the mechanistic interference of interfering compounds that have similar electrochemical peaks (AA, UA, and DA), chemical structure (lactose, mannose, and galactose), or surface electrocatalytic effects (KCl and NaOH) with the glucose sensor by Ye et al. [91].

**Figure 6 polymers-13-03706-f006:**
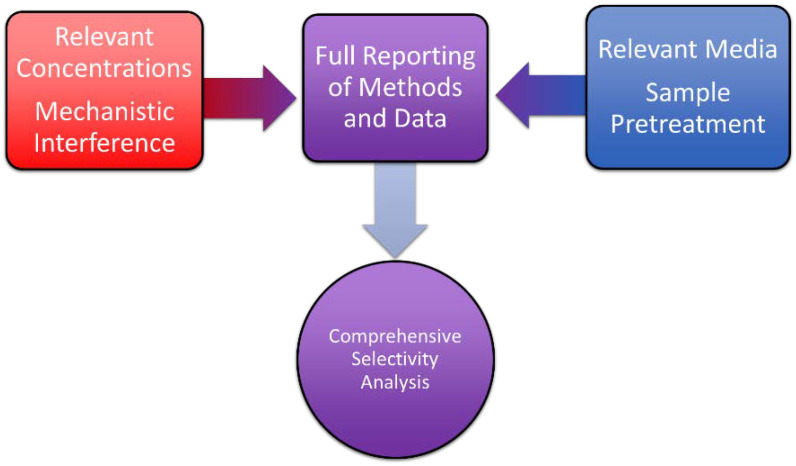
Graphic representation of how interferant experiments that take into account mechanistic interference and concentrations relevant to intended end-use in combination with complex media analysis that takes into account a media type relevant to the intended end-use and minimal sample pre-treatment results in a comprehensive analysis of selectivity for nanofiber sensors.

**Table 1 polymers-13-03706-t001:** Eleven nanofiber sensor examples with reported lower limit of detection (LDL) and precision or relative standard deviation (RSD). LDL indicates the general sensitivity of the sensor. RSD is chosen to show the general consistency of the measured sample. The # will consistently be used for Table 2 and Table 3.

#	General Material	Nanofiber Material	Analyte Tested	LDL	Precision (RSD)	Sensing Mech. ^+^	Ref.
1	Carbon	TiO_2_/CNF	Idarubicin hydrochloride	3 µM	2.40%	AD	[35]
2	Organic polymer/metal oxide	PLC/ZnO-NPs/CuO-NFs	Adenine, guanine	12.48 nM	2.3%	EC	[36]
			Guanine	1.25 nM	1.2%		
3	Metal oxide	CeBiO_x_	Acetaminophen	0.2 µM	0.49%	EC	[37]
4	Organic polymer	3D Cu_x_O-ZnO NP/PPyNF/RGO	Ascorbic acid	0.024 µM	0.67%	AD, EC	[38]
			Dopamine	0.012 µM	0.81%		
			Paracetamol	0.01 µM	0.95%		
			Tryptophan	0.016 µM	1.14%		
5	Organic polymer/metal oxide	PANI NF/PEG	DNA sequence	0.0038 pM	5.80%	AD and ASR	[39]
6	Peptide	GQD/PNF/GO	Hydrogen Peroxide	1.056 µM	N/R*	AD, EC	[40]
7	Carbon	NGQD/NCNF	Nitrite	3 µM	4.27%	EC	[41]
8	Organic polymer/metal oxide	CuO/PANI NF	H_2_O_2_	0.110 µM	N/R*	AD, EC	[42]
			Glucose	0.45 µM	N/R		
9	Carbon/metal/organic polymer	PMB-Cu-NF/ACF	Creatinine	0.2 ng/mL	1–2%	EC, AD, and ASR	[43]
10	Metal oxide	SnO_2_	Atrazine	0.9 zM	2.5%	AD and ASR	[44]
11	Peptide	PNF	Breast cancer stem-like cells	6 cells/mL	Within 10%	ASR	[45]

Key: N/R, not reported; N/R*, accuracy was reported but not precision. Nanofiber material abbreviations defined in Abbreviations: Back Matter. ^+^ Sensing mechanism: EC, electrocatalytic; AD, adsorption; ASR, analyte-specific recognition.

## Data Availability

Not applicable.

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
