# Peer review of "Perspective on Nanofiber Electrochemical Sensors: Design of Relative Selectivity Experiments"

_polymers, 2021, doi:10.3390/polym13213706_

Round 1

Reviewer 1 Report

The topic of this review is overall interesting. The authors focus on nanofiber-based sensors and try to provide the guidelines for selectivity experiments. However, after reading the manuscript, it presents the feeling of just piling up some recent literature without deeper and comprehensive analysis and discussion. The authors do not elucidate the key question of how various nanofibers enable the sensors to give their improved performances. It seems that many other materials could have the same function. The authors do not give a good answer for the key question “why is nanofiber special”. It is suggested to reject the manuscript for Polymers, which has high standards. The specific comments are listed below.

  1. For Table 1, the column labeled “Nanofiber Material”, it is suggested to give the specific material used, instead of the general terms like “carbon, metal oxide, organic polymer”.
  2. Page 4, between line 92 and 109, it is stated in the text that “Five additional examples of electrocatalytic sensing modalities are detailed:”, however, these seem to be just very brief description, instead of detailed analysis. The authors do not elucidate the nanofibers’ roles in these applications. The authors should elaborate more on these examples.
  3. Between line 128 and 143, analysis of these examples is very brief, not insightful. Moreover, line 128 states “Five additional examples of adsorptive mechanisms are detailed”, but actually only four examples are briefly listed.
  4. It lacks good analysis of Figure 1, 2 and 4, which are cited directly from the corresponding papers.
  5. Subscript errors and other typo errors are found through the manuscript. The authors should check carefully.
  6. Format of references is not consistent. 

Author Response

Please see our responses below or in the attached file.

Reviewer #1 Comments and Suggestions for Authors

The topic of this review is overall interesting. The authors focus on nanofiber-based sensors and try to provide the guidelines for selectivity experiments. However, after reading the manuscript, it presents the feeling of just piling up some recent literature without deeper and comprehensive analysis and discussion. The authors do not elucidate the key question of how various nanofibers enable the sensors to give their improved performances. It seems that many other materials could have the same function. The authors do not give a good answer for the key question “why is nanofiber special”. It is suggested to reject the manuscript for Polymers, which has high standards. The specific comments are listed below.

We appreciate and agree with this assessment. However, we are excited with the opportunity to provide a deeper and comprehensive analysis and discussion to the current manuscript framework. Based on your comments below, and the other reviewers, we were able to make a more detailed analysis of the nanofibers. While our focus for this paper was centered around the evaluation of the selectivity of nanofibers, we value your perspective that more information is needed on “why is nanofiber special”. We integrated more of detection mechanism and improvements nanofibers provide in sensing modalities.

  1. For Table 1, the column labeled “Nanofiber Material”, it is suggested to give the specific material used, instead of the general terms like “carbon, metal oxide, organic polymer”.

We added specific materials to Table 1 as an additional column. This new column added many acronyms. Instead of cluttering the footnote of this table, we placed the definition of these acronyms in a new abbreviation section located near the end of the manuscript.

  1. Page 4, between line 92 and 109, it is stated in the text that “Five additional examples of electrocatalytic sensing modalities are detailed:”, however, these seem to be just very brief description, instead of detailed analysis. The authors do not elucidate the nanofibers’ roles in these applications. The authors should elaborate more on these examples.

We agree that the summary for electrocatalytic sensing was too brief for an in-depth and detailed analysis. We added additional and specific sensing mechanisms for each of the bullet points. Upon further examination of this section, we decided to remove example 5 because the nanofiber interactions therein are similar to example 1. We also added specific values for important sensor properties and how these properties compare to other sensors referenced by the example. We also discussed the specific roles of the nanofibers in each of the example sensors.

  1. Between line 128 and 143, analysis of these examples is very brief, not insightful. Moreover, line 128 states “Five additional examples of adsorptive mechanisms are detailed”, but actually only four examples are briefly listed.

We agree that the summary for adsorptive mechanism was too brief for an in-depth and detailed analysis. We added additional detail to these examples. We gave specific values for important sensor properties and how these properties compare to other sensors referenced by the example. We also discuss the specific role of the nanofiber in the example’s sensors. We also corrected the heading of this list to only list four examples. We also recognized that the summary of the examples listed in section 2.3 Analyte-Specific Recognition was too brief. We also added detail to this section and gave specific values for improved sensor properties.

  1. It lacks good analysis of Figure 1, 2 and 4, which are cited directly from the corresponding papers.

Thank you for pointing this out. We used this opportunity to add more detail to the descriptions of Figure 1, 2, 3, and 4 in the text. We also tied this to the specific sensing modality discussed in this section, which we believe will better transition to the other descriptions listed.

  1. Subscript errors and other typo errors are found through the manuscript. The authors should check carefully.

We recognize this error. We have found, working with the MDPI template in formatting, that the superscripts and subscripts get altered in universally checking the formatting. We hope we corrected all of these errors in the latest draft.

  1. Format of references is not consistent. 

We were confused by this comment as we use a citation manager that converted all of the references to a Sensors & Actuators format. We did find some of the text within the manuscript converts to a different font than the MDPI template, which is compounded by comment #5 above. As in if the references refresh, some of them change to a different font or size. We hope we corrected all of this in the latest draft without it autocorrecting. Also, in trying to investigate this comment, we did find two references that did not have volume numbers and these were added.

Reviewer 2 Report

The review focuses on selectivity issues in nanofiber-based sensing. Selectivity is an extremely important factor for an analytical sensing application in real-life applications. The main references analyzed are very recent and adequate. Therefore, the review is timely and very useful for the field.
I only have a few general suggestions that may help improve the manuscript, as well as a very minor point.
- As a general comment, the review would benefit from adding summary figures, in visual form, that gathered at a glance the main points of the review. For instance: 1) a general visual figure of the different response mechanisms of nanofiber sensing; 2) a general visual figure of the types of selectivity tests, and its relation with the sensing mechanism. Such figures will help the reader with following the most relevant aspects of the review.
- In section 3.2, the authors could give more details and an overview classification of the different types of pre-treatment strategies that have been reported, and their correlation with the final application. This is, instead of just listing the examples in table 3 and describing a few of them, the authors can try to rationalize the different strategies.
- The first part of the introduction (lines 28-38) is a bit confusing, because nanomaterials can be used in many other applications, not only sensing. I got the feeling that the text moves excessively straightforwardly into the description of sensing nanomaterials. Therefore, it would be much more reader-friendly that the text first mentioned a general overview of nanomaterials, and then nanomaterials focusing on sensing applications.
- The acronym pMB is not defined in the text.

Author Response

Please see our responses below or in the attached file.

Reviewer 2: Comments and Suggestions for Authors

The review focuses on selectivity issues in nanofiber-based sensing. Selectivity is an extremely important factor for an analytical sensing application in real-life applications. The main references analyzed are very recent and adequate. Therefore, the review is timely and very useful for the field.

Thank you for this positive assessment of our work. We hope our changes only strengthen this assessment and usefulness.

I only have a few general suggestions that may help improve the manuscript, as well as a very minor point.
- As a general comment, the review would benefit from adding summary figures, in visual form, that gathered at a glance the main points of the review. For instance: 1) a general visual figure of the different response mechanisms of nanofiber sensing; 2) a general visual figure of the types of selectivity tests, and its relation with the sensing mechanism. Such figures will help the reader with following the most relevant aspects of the review.

Thank you for this suggestion. We decided to recycle the abstract figure for an introduction figure in the sensing mechanisms. We thought this figure does a great job summarizing the three different sensing modalities we describe. We had more difficulties thinking of a general visual figure of the types of selectivity tests. We created a new section titled “Future Perspectives” with a small bubble chart that may suffice. We are open to other suggestions if something else seems appropriate.

- In section 3.2, the authors could give more details and an overview classification of the different types of pre-treatment strategies that have been reported, and their correlation with the final application. This is, instead of just listing the examples in table 3 and describing a few of them, the authors can try to rationalize the different strategies.

Thank you for this great suggestion. We listed four distinct pre-treatment strategies and detailed why they are chosen and how they influence the aforementioned detection mechanism. We think this change was the most impactful change we made and added another layer to the manuscript.

- The first part of the introduction (lines 28-38) is a bit confusing, because nanomaterials can be used in many other applications, not only sensing. I got the feeling that the text moves excessively straightforwardly into the description of sensing nanomaterials. Therefore, it would be much more reader-friendly that the text first mentioned a general overview of nanomaterials, and then nanomaterials focusing on sensing applications.

Thank you for this suggestion. We added a new introduction paragraph giving a more general overview of nanofiber materials and the general uses of them. We also made some minor changes to the second paragraph to integrate with this new introduction outline.

- The acronym pMB is not defined in the text.

Thank you for this comment. We defined pMB in this latest draft. We also added many more acronyms based on the details added to satisfy the other reviewers. Because of these additional acronyms, we added an “abbreviation” section at the end of this draft.

Reviewer 3 Report

The authors intend to provide The goal a guidelines for acceptable nanofiber sensor selectivity experiments. Electrocatalytic, adsorptive, and analyte-specific recognition mechanisms were considered.
The review manuscript is within the scope of Polymers. However, before publication some major concerns must be addressed namely:

  1. The introduction should clearly focus the target of the audience and the timeframe that the authors intend to cover. Morever, why is necessary a review in this area? Who will benefit from this perspective? Young researchers? Senior researchers?
  2. Avoid the use of words such as "better" since it is not very accurate.
  3. Future perspectives must be approached and a critical analysis of the available literature must be also clearly introduced.
  4. Conclusions must be focused on the references and tables introduced through the manuscript and not so generic.
  5. The paper as presented is more a rational analysis than a review manuscript.

Author Response

Please see our responses below and in the attached file.

Reviewer 3: Comments and Suggestions for Authors

The authors intend to provide The goal a guidelines for acceptable nanofiber sensor selectivity experiments. Electrocatalytic, adsorptive, and analyte-specific recognition mechanisms were considered.
The review manuscript is within the scope of Polymers.

Thank you for this brief summary of our article and the suggested concerns listed below. We did our best to address all of your concerns.

However, before publication some major concerns must be addressed namely:

  1. The introduction should clearly focus the target of the audience and the timeframe that the authors intend to cover. Morever, why is necessary a review in this area? Who will benefit from this perspective? Young researchers? Senior researchers?

Thank you for this suggestion. We added the following sentence to the last paragraph of the introduction “Descriptions of sensible design of selectivity experiments, in the context of sensing mechanism, would be particularly useful to junior investigators designing their first sensor experiments.” We also added a similar statement to the abstract to make it very clear of the intended target audience

  1. Avoid the use of words such as "better" since it is not very accurate.

We completely agree with this suggestion. We removed all reference to the qualitivative words such as “better”, “very”, and when not substantiated with data, “more”.

  1. Future perspectives must be approached and a critical analysis of the available literature must be also clearly introduced.

Thank you for this suggestion. A Future Perspective section was added that we believe clarifies how the strategies described in the body should be applied to future experiments.

  1. Conclusions must be focused on the references and tables introduced through the manuscript and not so generic.

Thank you for this. We have added more detail to the conclusion in the form of references to tables and figures in the text.

  1. The paper as presented is more a rational analysis than a review manuscript.

We appreciate this summary of the text. We did not apply any meta-analysis or rational analysis, which places this closer to a review manuscript versus a rational analysis; however, we believe this article should be better classified as a perspective. In response to this and other reviewers’ comments, we have added significant detail to many sections of the manuscript, especially to the examples provided for sections 2.1, 2.2, 2.3, and 3.2. We also added more detail to the in-text descriptions of the figures, added more figures, and adjusted the introduction and conclusion. We hope the added qualitative literature assessment better support the manuscript being classified either as a perspective or review manuscript

Reviewer 4 Report

This review is about interesting and relevant topic, namely perspectives on nanofibers use for electrochemical sensors. It is known that due to the high surface area to volume ratio as well as high porosity of nanofibers they greatly improves such sensors parameters as the sensitivity, speed, and accuracy. Thanks to this review, you can efficiently plan your experiment about sensor selectivity with considerations for electrocatalytic, adsorptive, and analyte-specific recognition mechanisms. Electrochemical sensors are good alternative for sensors with mechanistic class-recognition nature, because selectivity problems can be solved by the thoughtful inclusion of relevant controls, the use of well-designed real sample analyses, and the inclusion of analyte-specific interactions. These comments will help to improve the quality of work:

  • A few typos in the manuscript. I recommend checking the contents of Table 2, namely lines 2 and 7. Pay special attention to the formulas of chemical compounds.

The literature review covers 70 sources, most of which are from 2020-2021, so the review is relevant and generalizing. Figures are sufficiently informative and of high quality. The review is sufficiently structured and has obvious conclusions. I think, that the manuscript can be published after correction according this minor remark.

Author Response

Please see our responses below and in the attached file.

Reviewer 4: Comments and Suggestions for Authors

This review is about interesting and relevant topic, namely perspectives on nanofibers use for electrochemical sensors. It is known that due to the high surface area to volume ratio as well as high porosity of nanofibers they greatly improves such sensors parameters as the sensitivity, speed, and accuracy. Thanks to this review, you can efficiently plan your experiment about sensor selectivity with considerations for electrocatalytic, adsorptive, and analyte-specific recognition mechanisms. Electrochemical sensors are good alternative for sensors with mechanistic class-recognition nature, because selectivity problems can be solved by the thoughtful inclusion of relevant controls, the use of well-designed real sample analyses, and the inclusion of analyte-specific interactions.

Thank you for this assessment.

These comments will help to improve the quality of work:

  • A few typos in the manuscript. I recommend checking the contents of Table 2, namely lines 2 and 7. Pay special attention to the formulas of chemical compounds.]

We recognize this error. We have found, working with the MDPI template in formatting, that the superscripts and subscripts get altered in universally checking the formatting. We hope we corrected all of these errors in the latest draft, but we will be extra conscious of these errors at the copy-editing phase if the manuscript gets accepted.

The literature review covers 70 sources, most of which are from 2020-2021, so the review is relevant and generalizing. Figures are sufficiently informative and of high quality. The review is sufficiently structured and has obvious conclusions. I think, that the manuscript can be published after correction according this minor remark.

Thank you for these comments and your support. Responding to other reviewers, we added several other articles.  

Round 2

Reviewer 1 Report

The revised manuscript has been improved and is thus suggested to be accepted.

Author Response

Thank you for this positive assessment. 

Reviewer 3 Report

I beleive the manuscript was much improved. I would reccomend publication after minor revision.

Author Response

Thank you for this positive assessment.